# Three-Dimensional Printable Photocurable Elastomer Composed of Hydroxyethyl Acrylate and Hydroxy Fatty Acid Derived from Waste Cooking Oil: An Innovative Strategy for Sustainable, Highly Flexible Resin Development

**DOI:** 10.3390/molecules30194000

**Published:** 2025-10-06

**Authors:** Fangping Shen, Chuanyang Tang, Yang Yang, Guangzhi Qin, Minghui Li, Haitian Jiang, Mengyao Wu, Shuoping Chen

**Affiliations:** College of Materials Science and Engineering, Guilin University of Technology, Guilin 541004, China; fangpingshen_221@163.com (F.S.); 2120230405@glut.edu.cn (C.T.); 1020230205@glut.edu.cn (Y.Y.); 1020230210@glut.edu.cn (G.Q.); 2120230354@glut.edu.cn (M.L.); 1020240199@glut.edu.cn (H.J.); 2120240427@glut.edu.cn (M.W.)

**Keywords:** 3D printing, photocurable elastomer, waste cooking oil, hydroxyethyl acrylate, hydroxy fatty acid

## Abstract

Waste cooking oil (WCO), a significant urban waste stream, presents untapped potential for synthesizing high-value materials. This study introduces an innovative “epoxidation-hydrolysis-blending” strategy to conveniently transform WCO into a highly flexible, photocurable elastomer suitable for 3D printing. Initially, WCO is converted into WCO-based hydroxy fatty acids (WHFA) via epoxidation and hydrolysis, yielding linear chains functionalized with multiple hydrogen-bonding sites. Subsequently, blending WHFA with hydroxyethyl acrylate (HEA) yields a novel photocurable WHFA/HEA elastomer. This elastomer exhibits excellent dimensional accuracy during vat photopolymerization 3D printing. Within the WHFA/HEA system, WHFA acts as a dual-functional modifier: its flexible alkyl chains enhance conformational freedom through plasticization while serving as dynamic hydrogen-bonding cross-linking sites that synergize with HEA chains to achieve unprecedented flexibility via reversible bond reconfiguration. Mechanical testing reveals that the optimized WHFA/HEA elastomer (mass ratio 1:3) exhibits ultra-high flexibility, with an elongation at break of 1184.66% (surpassing pure HEA by 360%). Furthermore, the elastomer demonstrates significant weldability (44.23% elongation retention after 12 h at 25 °C), physical reprocessability (7.60% elongation retention after two cycles), pressure-sensitive adhesion (glass interface adhesion toughness: 32.60 J/m^2^), and notable biodegradability (14.35% mass loss after 30-day soil burial). These properties indicate broad application potential in flexible electronics, biomedical scaffolds, and related fields. This research not only pioneers a low-cost route to multifunctional photocurable 3D printing materials but also provides a novel, sustainable solution for the high-value valorization of waste cooking oil.

## 1. Introduction

Vat photopolymerization (VP) 3D printing is a rapidly developing additive manufacturing technology, recognized for its high efficiency, fast curing, and excellent surface quality [1,2,3,4]. The recent introduction of low-cost LCD-based printers has further expanded its accessibility [5,6]. However, the broad application of VP remains limited by the dependency on petroleum-derived 3D printable photocurable resins, which are both environmentally unsustainable and costly [7,8,9,10,11,12]. There is a growing urgency to develop sustainable alternatives that support a circular economy, given the environmental impact and finite nature of fossil resources [13,14]. Several plant oil-derived resins, such as acrylated rapeseed oil-based resins [15] and palm oil-derived acrylamide monomers [16], have been proposed as sustainable alternatives to commercially available petroleum-based printing materials. These bio-based 3D printable consumables demonstrate tunable properties, including flexibility and shape memory capability. However, the use of edible plant oils as raw materials for 3D printing resins raises concerns regarding competition with food supplies, potentially exacerbating food security issues. Additionally, geopolitical and market fluctuations have led to volatile pricing and uncertain supply chains for plant oils [17,18]. In this context, waste cooking oil (WCO), a byproduct of food processing, presents a promising alternative for developing cost-effective and sustainable photocurable resins [19,20,21].

The valorization of WCO not only decreases petroleum dependence but also mitigates environmental and public health risks linked to improper disposal [22,23,24,25]. Structurally similar to edible oils, WCO consists mainly of triglycerides and free fatty acids, making it a suitable low-cost feedstock [26,27]. Currently, WCO is primarily recycled into biodiesel [28,29], with smaller applications in biolubricants [30,31,32], biosurfactants [33,34,35], and bioplasticizers [36,37]. Most of these pathways yield low added value products (typically <20%) and require large-scale operations to be profitable. Therefore, there is a critical need for innovative valorization routes that offer higher value and require lower capital investment [38,39,40,41,42,43]. In our previous studies, we have shown that WCO can be chemically modified through functionalization of unsaturated bonds and ester groups, transforming it into higher-value products such as solid alcohol [44], waxes [45,46], coatings [47], adhesives [48], and 3D printable photocurable resins [49,50,51]. Photocurable resins are particularly attractive due to their high added value (>100%) and functional versatility, supporting properties like high elasticity, shape memory, self-healing, pressure-sensitive adhesion, and biodegradability [52]. Compared to petroleum-based resins, WCO-derived photopolymers can reduce costs, enhance lifecycle sustainability, avoid food competition, and promote waste upcycling, meeting the growing demand for sustainable 3D printable consumables.

In our previous studies, we developed three stepwise strategies to synthesize WCO-derived photocurable resins with increasing performance (Figure 1). The initial approach (strategy 1 in Figure 1) utilized epoxidized WCO (E-WCO) as a plasticizer blended into a triethylene glycol dimethacrylate (TEGDMA) matrix, which moderately enhanced ductility, increasing elongation at break from 4.39% to 32.93% [49]. While simple, this method offered limited flexibility. The second strategy (strategy 2 in Figure 1) employed a two-step epoxidation-esterification process to synthesize photocurable monomers like methacrylated epoxy waste oil (EWOMA). By copolymerizing EWOMA with flexible acrylates, tunable elastomers were achieved with elongations up to 162.0% [48]. However, the inherent triglyceride branching of EWOMA resulted in densely crosslinked networks that restricted chain mobility, limiting extensibility to below 300%. To overcome this structural constraint, a third strategy (strategy 3 in Figure 1) introduced a linear monomer, WCO-based methacrylate fatty acid ethyl ester (WMFAEE), synthesized through transesterification, epoxidation, and ring-opening esterification. When copolymerized with hydroxypropyl acrylate (HPA), the WMFAEE-based elastomer exhibited significantly improved flexibility (645.09% elongation), facilitated by linear chain topology and dynamic hydrogen bonding. This advanced material also demonstrated multifunctional properties, including self-healing, weldability/reprocessability, pressure-sensitive adhesion, shape memory, and biodegradability [52].

Inspired by our early success with E-WCO plasticization and recognizing the limitations of triglyceride-derived networks, we sought to develop a more efficient route to convert WCO into high-performance photocurable elastomers. While E-WCO served as an effective plasticizer, its branched structure and limited hydrogen-bonding sites hindered the formation of robust yet dynamic networks. Herein, we propose an innovative “epoxidation-hydrolysis-blending” strategy to address these challenges (Figure 2). This approach involves the following steps: (1) Epoxidation-Hydrolysis of WCO: Conversion of WCO into WCO-based hydroxy fatty acids (WHFAs) with linear configurations and abundant hydrogen bond sites (average of two hydroxyl groups and one carboxyl group per molecule); (2) Network Design: Blending WHFAs with hydroxyethyl acrylate (HEA) to create a photocurable system where WHFAs act as both flexibilizers (via aliphatic chains) and dynamic crosslinkers (via hydrogen bonds). The resultant WHFA/HEA elastomer achieves an unprecedented elongation at break exceeding 1000%, surpassing all prior WCO-derived resins while retaining reprocessability and biodegradability. We systematically investigate the roles of WHFA content, hydrogen-bonding density, and network topology on mechanical properties, adhesion, and environmental stability, offering guidelines for designing sustainable, high-performance 3D printing materials.

## 2. Results and Discussion

### 2.1. Spectroscopic Analysis of WHFA Based on WCO

Figure 3a presents a comparative analysis of the ^1^H NMR spectra for WCO, E-WCO, and WHFA derived from WCO. As illustrated, the spectrum of raw WCO (primarily triglycerides) exhibited two split quadruplets in the region δ 4.1–4.4 ppm, corresponding to the methylene protons (−CH_2_−O−CO−R) of the glycerol backbone, and a triplet at δ 5.0–5.1 ppm, attributed to the methine proton (−CH−O−CO−R). Additionally, characteristic signals for unsaturated double bonds (−CH=CH−) appeared at δ 5.2–5.4 ppm [53,54]. Following epoxidation to yield E-WCO, the proton signals associated with the glycerol backbone remained observable. However, the unsaturated double bond signals at δ 5.2–5.4 ppm were entirely absent, while new characteristic signals for the epoxy group emerged in the region of δ 2.8–3.2 ppm, indicating successful conversion of the double bonds to epoxide groups [55].

In the spectrum of the hydrolyzed product (WHFA), the signals corresponding to the methylene and methine protons of the triglycerides were completely abolished. This confirmed the complete deconstruction of the branched glycerol backbone, resulting in linear fatty acids. Concurrently, the epoxy group signals at δ 2.8–3.2 ppm disappeared entirely. New signals appeared at δ 3.2–3.8 ppm, attributable to the methine protons adjacent to hydroxyl groups in a vicinal diol structure (−CH(OH)−CH(OH)−). This indicated that the hydrolysis process also facilitated the cleavage of the epoxide rings, forming dihydroxyl structures, resulting in a mixture of hydroxyl fatty acids predominantly including compounds like 9,10-dihydroxystearic acid [46].

Infrared (IR) spectroscopy further verified the progression of the epoxidation and hydrolysis reactions (Figure 3b). Following epoxidation, the characteristic absorption bands in raw WCO associated with C=C bonds significantly weakened or disappeared. These included the =C−H stretching vibration at 3006 cm^−1^ and the C=C stretching vibration at 1643 cm^−1^. Simultaneously, a distinct characteristic absorption band for the epoxy group appeared at 832 cm^−1^, confirming the successful synthesis of E-WCO. Subsequent hydrolysis of E-WCO resulted in the complete disappearance of the ester carbonyl (C=O) stretching vibration at 1747 cm^−1^. Concurrently, a new band appeared at 1709 cm^−1^, which is characteristic of the carbonyl stretching vibration in carboxylic acids. This confirmed the conversion of E-WCO into free fatty acids. Furthermore, the epoxy absorption band at 832 cm^−1^ vanished, while a new band assigned to the C−O stretching vibration of secondary alcohols appeared at 1105 cm^−1^. This series of spectral changes demonstrated the complete transformation of the epoxide groups into dihydroxyl structures, confirming the reaction pathway from epoxidation through hydrolysis [53,56].

### 2.2. Spectroscopic Analysis of WHFA/HEA Photocurable Elastomer

Structural transformations induced by photopolymerization in the WHFA/HEA system were systematically investigated using infrared (IR) spectroscopy, focusing on the optimal A2 formulation (WHFA/HEA mass ratio = 1:3). As depicted in Figure 4a, the liquid precursor exhibited characteristic absorption bands of unsaturated moieties. These included C=C stretching vibrations (1597 cm^−1^), =CH_2_ in-plane (1376 cm^−1^) and out-of-plane (986 cm^−1^) bending, as well as −CH= in-plane (1296 cm^−1^) and out-of-plane (814 cm^−1^) bending modes [57,58]. Post-photopolymerization, these bands were markedly attenuated or eliminated, indicating dominant polymerization of HEA’s C=C bonds. Quantitative analysis revealed a double bond conversion (DC) of 96.05%, confirming near-complete photopolymerization of HEA under UV irradiation.

Comparative IR analysis highlighted the role of WHFA in modulating polymer network interactions. The cured WHFA/HEA elastomer displayed a broad absorption band at 3178 cm^−1^, attributed to hydroxyl (−OH) stretching vibrations with strong intermolecular associations (Figure 4b). This feature, insignificant in pristine HEA, provided direct evidence for WHFA-induced hydrogen bond formation within the polymer matrix [59,60]. Such interactions suggest WHFA acts as a crosslinking agent, enhancing structural integrity through non-covalent bonding.

X-ray photoelectron spectroscopy (XPS) further validated these structural modifications (Figure 4c–d, Appendix A). Survey scans confirmed the presence of carbon and oxygen in both systems, while high-resolution C1s and O1s spectra revealed distinct binding energy shifts. In the WHFA-modified network, C−O bonds shifted from 286.65 eV to 286.45 eV, and C=O bonds transitioned from 289.00 eV to 289.30 eV in the C1s region. Concurrently, O1s spectra showed positive shifts for C=O (533.22 → 533.54 eV) and C−O (532.20 → 532.47 eV) environments. These systematic energy changes unambiguously demonstrate that WHFA was integrated into the HEA matrix and played a pivotal role in forming an extensive hydrogen-bonded architecture [61,62,63].

### 2.3. Migration Rate of WHFA Based on WCO

The migration behavior of WHFA as a small molecule within the HEA matrix was systematically investigated to assess its stability under various conditions. Migration tests were conducted on WHFA/HEA samples (A1–A4) at 135 °C with different heating durations, using E-WCO/HEA and SA/HEA resins as controls (Figure 5a). The results revealed a time-dependent increase in WHFA migration during the initial heating period. The migration reached equilibrium after 10 h as indicated by stabilized mass loss. A composition-dependent trend was observed, where samples with higher WHFA content exhibited proportionally greater migration rates. Specifically, the A2 sample (WHFA:HEA mass ratio = 1:3) demonstrated only 1.25% mass loss after 24 h at 135 °C. In comparison, under the same formulation, the migration rate of E-WCO reached 1.99%, while SA with smaller molecular weight exhibited a migration rate exceeding 4%. This discrepancy occurred because the hydroxyl and carboxyl groups in WHFA molecules formed stronger hydrogen bonds with the HEA polymer chains. This anchoring effect effectively reduced molecular mobility and enhanced thermal stability [64,65]. Additionally, WHFA exhibited significantly improved stability under milder conditions: no detectable mass loss occurred during extended storage at room temperature, and heating at 60 °C for 24 h resulted in merely 0.06% migration. These findings demonstrated the material’s suitability for long-term applications in environments below 100 °C, with minimal concerns regarding WHFA leaching (Figure 5b).

### 2.4. Thermal Analysis

The thermogravimetric analysis (TGA) results for the 3D printed WHFA/HEA composite products and pure HEA were presented in Figure 6a, Appendix A. It was observed that the initial decomposition temperature of pure HEA was 370 °C. The incorporation of WHFA slightly reduced the thermal stability of the WHFA/HEA system. Moreover, the decomposition temperature of the WHFA/HEA composites decreased with an increase in WHFA content. Nevertheless, the decomposition temperature of sample A2, which exhibited the best mechanical properties, still reached 331 °C, indicating that it could meet general usage requirements.

On the other hand, the differential scanning calorimetry (DSC) results revealed endothermic peaks within the temperature range of −30 °C to 0 °C for all samples, indicating their glass transition behavior (Figure 6b, Appendix A). Pure HEA exhibited the highest glass transition temperature (T_g_) of −4.9 °C. The incorporation of WHFA significantly reduced the T_g_ of the system, with the T_g_ decreasing progressively as the WHFA content increased. For instance, sample A2 showed a T_g_ of −18.2 °C.

This decrease in T_g_ can be attributed to the flexible aliphatic chains of WHFA, which act as molecular spacers and lubricants within the HEA polymer matrix. These disruptions to intermolecular interactions increased the free volume and facilitated segmental motion, demonstrating a clear plasticizing effect [66,67]. The results suggest that the WHFA/HEA composites are capable of exhibiting high elasticity at room temperature, making them suitable for use as high-performance elastomers. Thus, WHFA functions as an effective plasticizer by weakening the intermolecular forces between HEA chains, enhancing chain mobility, and improving the overall flexibility of the material.

### 2.5. 3D Printing Behavior

To verify the applicability of the WCO-based WHFA/HEA photocurable elastomer in photocurable 3D printing, the penetration depth (D_p_) and critical exposure energy (E_c_) of WHFA/HEA photocurable elastomers with different WHFA contents were tested using 405 nm purple light as light source. Pure HEA and E-WCO/HEA composites were used as controls, and the results are presented in Figure 7a. It was evident that pure HEA exhibited a relatively low D_p_ value of 0.128 mm. When E-WCO was employed as a plasticizer, the D_p_ value of the system increased to 0.148 mm. However, when WHFA was introduced, the D_p_ value continuously decreased with the addition of WHFA. For example, the D_p_ value of sample A2 was only 0.065 mm, and its E_c_ was 6.75 mJ/cm^2^ (Figure 7a, Appendix A). The D_p_ parameter reflects the material’s sensitivity to variations in light output. A lower D_p_ value indicates higher tolerance to fluctuations in exposure time or light source power. This enhanced stability improves printing accuracy and commercial viability. In contrast, materials with higher D_p_ values may achieve faster printing speeds but are more sensitive to deviations in light output, which complicates precise control [8,68]. Compared with other reported materials (Table 1), the WHFA/HEA photocurable elastomer exhibited a lower D_p_ value than commercial resins (0.314 mm). This implies easier control of printing conditions, improved fault tolerance, and a higher likelihood of producing high-quality products.

Actual printing tests further validated its ability to achieve high-precision printing. When the layer thickness was set at 50 μm, the Z-axis dimensional deviation of the samples molded from WHFA/HEA resin was as low as 0.12% (Figure 7b), and the in-plane geometric error was as low as 0.047% (Figure 7c). These data demonstrate that the WHFA/HEA resin successfully synergized high resolution, dimensional fidelity, and surface quality. This combination provides strong support for its use in advanced 3D printing applications.

### 2.6. Mechanical Properties

To investigate the influence of WHFA introduction on the mechanical properties of the HEA polymer system, the mechanical properties of WHFA/HEA photocurable elastomers with varying WHFA contents were tested. Pure HEA and E-WCO/HEA polymers were used as controls. As depicted in Figure 8a, as well as Appendix A, the pure HEA resin was already a material with a certain degree of flexibility, exhibiting a tensile strength of 0.42 MPa and an elongation at break of 256.6%. After the introduction of WHFA, the mechanical properties of the material, especially in terms of flexibility and stretchability, were significantly enhanced. Moreover, as the amount of WHFA increased, the mechanical properties initially rose and then declined. The A2 sample, obtained when the mass ratio of WHFA to HEA was 1:3, demonstrated the optimal mechanical properties. Its tensile strength reached 0.96 MPa, which was 2.3 times that of pure HEA, and its elongation at break soared to 1184.66%, 4.6 times that of pure HEA.

It was noteworthy that although E-WCO could also improve the flexibility of the HEA system, the extent of improvement was far less than that of WHFA. The elongation at break of the WHFA/HEA elastomer (A2 sample) (1184.66%) was nearly twice that of the E-WCO/HEA resin with the same proportion (595.35%). On the other hand, it was found that if stearic acid (SA), which lacks hydroxyl groups, was used to replace WHFA, not only did it fail to enhance the mechanical properties of the system, but it also reduced the flexibility. The SA/HEA had a tensile strength of only 0.22 MPa and an elongation at break of only 69.71%, less than 6% of that of the WHFA/HEA elastomer.

This indicated that, unlike E-WCO, WHFA did not act merely as a plasticizer by increasing conformational freedom of HEA chains through its flexible alkyl chains. Its functionality extended further. WHFA molecules possessed multiple hydrogen bonding sites, including at least two hydroxyl groups and one carboxyl group per molecule. These groups enabled WHFA to form effective hydrogen-bonding crosslinking anchors. By forming a dynamic hydrogen bonding network with HEA, it enhanced the flexibility of the system. In contrast, the SA molecule had only one hydrogen bonding site (the carboxyl group) and could not form an effective hydrogen bonding cross linking network. Therefore, the introduction of SA without hydroxyl groups actually reduced the flexibility of the material.

Compared with pure HEA, the high elasticity of the WHFA/HEA elastomer was mainly attributed to the following factors. Pure HEA forms a linear polymer with closely packed chains and inter-chain hydrogen bonds, which restrict chain segment movement and limit flexibility. In contrast, the introduction of WHFA into WHFA/HEA had two effects. Firstly, the flexible long alkyl chains in WHFA had a certain plasticizing effect, expanding the conformational freedom of the polymer chains and allowing for significant elastic deformation. Secondly, as a hydrogen bonding cross linking anchor point, WHFA could synergize with adjacent HEA chains to form a dynamic hydrogen-bonding cross-linking network. This network enables large deformations while preventing fracture through continuous breaking and reformation of hydrogen bonds. The synergy of this dual mechanism (the plasticizing effect of flexible alkyl chains and the dynamic hydrogen bonding network) achieved unprecedented high flexibility in the waste oil resin system (Figure 8b).

Cyclic tensile tests further elucidated the dynamic mechanical behavior of the A2 elastomer. Under varying strain levels, hysteresis loops were observed across all five cycles (Figure 8c), with the dissipated energy demonstrating a strain-dependent increase from 0.038 MJ/m^3^ at 150% strain to 0.215 MJ/m^3^ at 750% strain (Figure 8d). This strain-responsive trend was attributed to the strain-induced disruption of hydrogen bonds within the polymeric network. At 600% strain, the first cycle exhibited the largest hysteresis loop, with a hysteresis loss ratio (HLR) reaching up to 46.66%, which subsequently decreased progressively in subsequent cycles, stabilizing at 26.67% after the fifth cycle. This phenomenon indicated that hydrogen bond recombination was initially delayed during the first deformation but became more efficient in later cycles, leading to reduced energy dissipation (Figure 8e–8f).

In comparison with petroleum-derived elastomers documented in the literature (Figure 8g and Table 2), the WHFA/HEA resin developed in this study exhibited significantly enhanced flexibility. For instance, relative to hydroxyl-terminated polybutadiene (HTPB)/hydroxypropyl methacrylate (HPMA) copolymer, which demonstrated an elongation at break of 246.1%, the WHFA/HEA system achieved an approximately 3.8-fold increase in elongation [74]. The WHFA/HEA resin also outperformed several previously reported plant oil-based elastomers; its elongation at break was 139% that of the palm oil-derived elastomer (851% elongation) reported by Wu et al. [16]. However, the tensile strength of the WHFA/HEA resin (0.962 MPa) was lower than that of most petroleum-based resins. Nevertheless, a strength around 1 MPa is generally sufficient for daily applications.

Moreover, due to the incorporation of waste cooking oil-derived WHFA, the WHFA/HEA resin featured a relatively low production cost ($13.09/kg, Appendix A), substantially below that of commercial flexible photocurable resins, which often exceed $50/kg. This cost-effectiveness, combined with its ecological benefits, positions the WHFA/HEA resin as a promising sustainable alternative to petroleum-based photocurable materials, particularly suitable for applications requiring large deformations. It is anticipated that, with appropriate design and processing, this resin could be employed in flexible wearable devices, electronic packaging, biological scaffolds, and deformable toys. For example, a small sword 3D printed from the A2 elastomer underwent considerable elastic deformation, rapidly recovered upon load removal, and endured repeated usage cycles without failure (Figure 8h).

Additionally, the stretchability of the WHFA/HEA photocurable elastomer exceeded that of earlier WCO-based elastomers (Figure 8g and Table 2). For instance, compared with the WCO based WMFAEE-HPA elastomer prepared using the “transesterification & epoxidation & ring-opening esterification & blending” strategy reported by us recently (with an elongation at break of 645.09%) [52], the elongation at break of the WHFA/HEA increased by 1.84 times. Moreover, the preparation process of WHFA/HEA (requiring only three steps: epoxidation, hydrolysis, and blending) was more convenient. However, compared with the WMFAEE-HPA elastomer, the biomass content of the WHFA/HEA photocurable elastomer (about 24%) was lower than that of the WMFAEE-HPA elastomer (38%), and it could not integrate self-healing and shape memory properties like the latter.

### 2.7. Welding and Physical Reprocessing Properties

In addition to being amenable to 3D printing, the WHFA/HEA elastomer could also be assembled through customized processing techniques such as room-temperature welding, thereby expanding its applicability in personalized manufacturing. To evaluate the weldability of the A2 elastomer, a cut sample strip was prepared with the fractured ends overlapped by 5 mm. A constant pressure was applied to the overlapping region using a 500 g weight at 25 °C for 12 h. The weldability was subsequently observed and assessed, as illustrated in Figure 9a. The results showed that the specimens formed a stable connection. The welded joints retained 21.24% of the original tensile strength and 44.23% of the elongation at break (Figure 9b, as well as Appendix A).

Notably, under identical conditions, pure HEA polymer failed to form a stable weld. This contrast confirmed that WHFA, serving as hydrogen-bonding anchors, played a critical role in enabling interfacial bonding. It could be inferred that the welding capability was primarily attributed to WHFA-mediated hydrogen-bond recombination at the interface, complemented by the mutual diffusion of long alkyl chains [77,78]. However, defects at the interface and an incomplete hydrogen-bonding network limited the full recovery of mechanical properties.

The welding behavior observed indicated that, due to the presence of WHFA which provided abundant hydrogen-bonding anchors, the WHFA/HEA elastomers were capable of bonding under specific conditions. It could be anticipated that waste WHFA/HEA elastomer powder might be reshaped through a similar mechanism. Experiments confirmed that the waste elastomer powder could be rapidly reprocessed via hot-pressing into bulk materials with recoverable mechanical properties. This reprocessability enhanced manufacturing flexibility and significantly improved the material’s reuse value in post-consumer waste streams, aligning with circular economy principles [79,80]. To verify this property, the A2 elastomer was ground into fine powder and then hot-pressed at 180 °C under a pressure of 4 MPa for half an hour to assess its reprocessability (Figure 10a). After two reprocessing cycles, the material retained 12.60% of its original tensile strength and 7.60% of its elongation (Figure 10b, as well as Appendix A). The deterioration of its mechanical properties might be attributed to the accumulation of defects during the reprocessing, such as voids and chain breakage. Nevertheless, it could still be used for general purposes. FT-IR analysis showed no significant changes in chemical groups after reprocessing, indicating that the process involved physical rather than chemical changes (Figure 10c).

### 2.8. Pressure Sensitive Adhesive Properties

In addition to significantly enhancing the flexibility of the material, the introduction of WHFA also endowed the WHFA/HEA photocurable elastomers with certain pressure-sensitive adhesive (PSA) properties. Dynamic mechanical analysis (DMA) revealed distinct viscoelastic properties for pure HEA within the temperature range of 15–40 °C. Its storage modulus (G′) was markedly higher than its loss modulus (G″), and the loss factor (tanδ) remained well below 1. This indicated that elastic behavior dominated, rendering pure HEA impractical for use as a PSA. In contrast, the WHFA/HEA elastomer exhibited a relatively balanced viscoelasticity at room temperature (15–40 °C), where G′ and G″ gradually approached each other, and tanδ was closer to 1 compared with pure HEA. This suggested a tendency toward equilibrium between molecular diffusion (viscosity) and energy dissipation (elasticity) [81,82]. This suggested that WHFA improved the mobility of HEA molecular chains, enhancing their diffusion, wetting capability, and cohesion via dynamic hydrogen-bonding networks. Additionally, interactions with adherend surfaces contributed to the observed PSA properties (Figure 11a, Appendix A).

Actual PSA adhesion tests corroborated this finding. Although pure HEA, rich in hydrogen bonds, exhibited interfacial adhesion toughness below 15 J/m^2^ on various surfaces (with a maximum of 11.10 J/m^2^ on glass), it had little practical value. In contrast, the WHFA/HEA elastomer demonstrated higher interfacial adhesion toughness than pure HEA on common surfaces. Notably, its interfacial adhesion toughness on glass reached 32.60 J/m^2^, three times that of pure HEA, enabling its use as a PSA material for glass surfaces (Figure 11b, Appendix A). However, the PSA performance of the WHFA/HEA elastomer was inferior to that of other reported WCO-based elastomers [48,50,52]. This discrepancy might be because WHFA, being a small molecule, cannot form covalent bonds with HEA chains. This limitation restricts its ability to finely regulate viscoelasticity and achieve a closer balance between G′ and G″.

In combination with its 3D printable characteristic, the WHFA/HEA elastomer could adhere persistently to glass surfaces in vertical, inclined, or suspended states and undergo reversible elastic deformation on such surfaces (Figure 11c). Furthermore, the photocurable nature of the WHFA/HEA elastomer enables its convenient fabrication into PSA tapes or other products by manual coating. For example, liquid resin could be coated onto a plastic substrate, covered with a transparent release film to eliminate air bubbles, and cured under 405 nm UV irradiation for 120 s to obtain a PSA product adhering to glass surfaces (Figure 11d).

### 2.9. Biodegradability

To mitigate environmental pollution from waste plastics and promote harmonization between materials and the ecosystem, it is essential to enhance the biodegradability of materials after service life via pathways such as soil burial [83,84]. Based on our previous studies, polymers derived from WCO generally exhibit favorable biodegradability [48,51,52]. Similarly, WHFA—synthesized from WCO—also demonstrated good biodegradable characteristics. Incorporating WHFA into the HEA polymer not only reduced the consumption of HEA and improved mechanical properties but also enhanced biodegradability, thereby reducing environmental pollution.

A 30-day soil burial experiment was conducted to evaluate and compare the biodegradability of WHFA/HEA and pure HEA. As shown in Figure 12a and Appendix A, pure HEA displayed limited biodegradability, with a degradation rate of approximately 2% within the first 5 days, reaching only 5.77% by day 30. In contrast, the WHFA/HEA elastomer (A2 sample) exhibited significantly improved biodegradation performance: its degradation rate reached 5.94% after 5 days and increased to 14.35% by day 30, demonstrating a consistent upward trend.

Microscopic morphological investigations revealed distinct structural changes between pristine and degraded samples. The original WHFA/HEA elastomer (Figure 12b) exhibited a smooth, homogeneous surface, whereas the degraded counterpart (Figure 12c) displayed pronounced roughness, microporosity, and microbial colonization—evidence of polymer matrix erosion mediated by soil microorganisms. These findings confirm that the integration of WHFA significantly improves biodegradability, reducing environmental burden and aligning with green chemistry principles. The WHFA/HEA elastomer thus demonstrates substantial promise as an eco-friendly material for sustainable applications.

## 3. Materials and Methods

### 3.1. Materials

The waste cooking oil (WCO) utilized in this study was sourced from the dining facilities at Guilin University of Technology, located in Guilin, China. The iodine value of the WCO, a measure of unsaturation defined as the grams of iodine absorbed by 100 g of oil, was assessed to be 108.2 g I_2_/100 g through standard titration techniques. Gas chromatography-mass spectrometry (GC-MS) analysis of the fatty acid composition further revealed that oleic acid (C 18:1) was the predominant fatty acid, accounting for 46.1% by weight of the total fatty acids (Appendix A). Hydrogen peroxide (H_2_O_2_, analytical grade, 30 wt% in H_2_O) and sulfuric acid (98%) were procured from Xiya Chemical Technology Co., Ltd., based in Linyi, China. Additional synthetic reagents, including sodium ethylate (20% w/w ethanol solution), anhydrous ethanol (99.9%), glacial acetic acid (99.5%), urea (99%), sodium bicarbonate (99.5%), hydroquinone (HQ, 99%), phenylbis(2,4,6-trimethylbenzoyl)phosphine oxide (Irgacure 819, 98%), *p*-dimethylaminobenzaldehyde (DMAB, 99%), stearic acid (SA, 98%), and hydroxyethyl Acrylate, (HEA, 97%), were all acquired from McLean Company, located in Shanghai, China.

### 3.2. Preparation of WHFA

The WHFA was synthesized from WCO via a sequential two-step strategy: epoxidation and hydrolysis reaction [46], as detailed below:

(1) Synthesis of E-WCO

The previous research findings demonstrated the feasibility of synthesizing epoxidized waste cooking oil (E-WCO) through the epoxidation method. The typical procedure involved mixing 1240 g of 30% H_2_O_2_, 320 g of acetic acid, 8 g of sulfuric acid, and 8 g of urea in an opaque container, followed by placing the mixture in an oven at 40 °C for 12 h to generate peracetic acid as the epoxidizing agent. Subsequently, 1000 g of WCO was added into a stirred glass reactor and heated to 40 °C, after which the prepared epoxidizing agent was gradually introduced into the reactor over 2 h. The reaction mixture was then heated to 70 °C and stirred for 3 h. Upon completion of the reaction, the mixture was allowed to stand for 12 h for phase separation. The upper oil layer was collected and subjected to a single wash with 5% NaHCO_3_ solution at 60 °C, followed by two washes with deionized water. Finally, the product was obtained as a clear yellow oily liquid (Appendix A), identified as E-WCO, after vacuum evaporation at 70 °C.

(2) Synthesis of WCO based hydroxy fatty acid (WHFA)

The E-WCO synthesized above was subjected to a ring-opening reaction in an alkaline hydrolysis system. A lye solution was prepared by dissolving 5.0 g of NaOH in 10 mL of deionized water. Under magnetic stirring, the lye solution was added dropwise to E-WCO, and the reaction was carried out at a constant temperature of 60 °C for 4 h. After the reaction, 13.2 g of a 10% HCl solution was added dropwise to the system to adjust the pH to neutral (pH = 7.0). The product was then centrifuged (8000 rpm, 10 min) and vacuum-dried (60 °C, 24 h). A clear, light-yellow transparent liquid was obtained (Appendix A), and after cooling, a near-white waxy solid was obtained, which was identified as WHFA.

### 3.3. Preparation of WHFA/HEA Photocurable Elastomers

WHFA and hydroxyethyl acrylate (HEA) were blended at the mass ratios shown in Table 3 at 60 °C to form a homogeneous pre-polymer. Subsequently, a photoinitiation system was added, which consisted of Irgacure 819 (photoinitiator) and DMAB (promoter) in a 1:1 mass ratio. The addition amount of the photoinitiation system was 3 wt% of the total mass of the mixed monomers. The mixture was stirred at 60 °C in the absence of light until homogeneous and then cooled to room temperature, yielding a light yellow transparent liquid WHFA/HEA resin. The resulting resins exhibited a room-temperature viscosity in the range of 20–50 mPa·s (Appendix A). Pure HEA, E-WCO/HEA and SA/HEA composites were used as control samples.

### 3.4. Molding of WHFA/HEA Photocurable Elastomers

The obtained liquid WHFA/HEA elastomers could be photocuring molded using two methods.

(1) 3D printing process

The 3D printing of WHFA/HEA elastomer was completed using a PhotonX LCD light curing 3D printer (Anycubic, Shenzhen, China). During the printing process, a 405 nm ultraviolet light source (irradiation intensity of 3.6 mW/cm^2^) was used for top-down layer-by-layer exposure and curing. The prepared WHFA/HEA liquid resin was injected into the printer’s resin tank, and a preset STL 3D model file was imported. The key printing parameters were optimized as follows: the bottom layer exposure time was 130 s (to ensure substrate adhesion), and the normal layer exposure time was 80 s (with a layer thickness of 50 μm). The final 3D printed products were yellow, smooth surfaced, soft, and tough solids (as shown in Figure 2). After cleaning the product surface with ethanol and drying in the air, they could be used for performance testing and other applications.

(2) Traditional photocuring coating

In the application of PSA, the liquid WCO-based elastomer could also be evenly applied onto substrates (such as metal foil, paper, cotton fabric, or polylactic acid plastic), followed by the placement of a transparent plastic release film to remove air bubbles. A 405 nm UV lamp was then used to irradiate the resins from the side of the transparent release film for 60 s, curing the resins and producing products with PSA properties. During use, the release film could be removed, allowing the product to adhere to specific glass surfaces.

The control samples, including SA/HEA, E-WCO/HEA, and pure HEA, were prepared using the same molding conditions as the WHFA/HEA elastomer.

### 3.5. Characterization

The physicochemical attributes and functional performance of the WCO-based photocurable elastomer synthesized in this study were comprehensively analyzed through standardized analytical methodologies established in prior publications. A full description of experimental procedures [47,48,49,50,51,52], encompassing precise instrumental configurations and additional analytical data, has been systematically compiled in Appendix A for transparency and reproducibility.

## 4. Conclusions

This study presents a groundbreaking strategy for transforming waste cooking oil (WCO) into a high-performance, photocurable resin compatible with 3D printing. The key innovation lies in converting WCO into WHFA—a linear, hydrogen bond-rich modifier—via an epoxidation-hydrolysis process. When blended with hydroxyethyl acrylate (HEA), the resulting WHFA/HEA resin enables high-precision light-curing 3D printing while exhibiting exceptional flexibility. WHFA serves a dual role: (1) its flexible alkyl chains enhance polymer chain mobility through plasticization, and (2) its hydrogen-bonding sites form a dynamic crosslinking network that enables large deformations while maintaining structural integrity via reversible bond reconfiguration. The optimized WHFA/HEA elastomer (1:3 mass ratio) achieves an unprecedented elongation at break of 1184.66%—4.6 times that of pure HEA—setting a new benchmark for WCO-derived photocurable resins.

Beyond exceptional flexibility, the WHFA/HEA system exhibits a multifunctional profile encompassing weldability, reprocessability, pressure-sensitive adhesion, biodegradability, and superior print fidelity. These properties make it a compelling candidate for sustainable additive manufacturing in diverse fields such as flexible electronics, adaptive soft robotics, environmentally friendly adhesives, smart packaging, wearable devices, and customizable toys or scaffolds. Building on our prior work, this study confirms that molecular-level engineering of WCO is key to designing cost-effective, high-performance 3D printing materials. By utilizing WHFA as a dynamic additive, we establish a scalable, high-value upcycling pathway for WCO, underscoring the pivotal role of tailored hydrogen-bonding networks in advanced polymer design.

The developed “epoxidation-hydrolysis-blending” approach represents a significant advance in WCO valorization. Unlike conventional biodiesel production, this method operates under mild conditions with minimal equipment, offering economic viability for resource-limited regions. Future research should focus on scaling WHFA synthesis, evaluating long-term environmental stability of printed components, and exploring WHFA’s utility in multi-material systems to broaden the impact of waste-derived functional polymers.

## Figures and Tables

**Figure 1 molecules-30-04000-f001:**
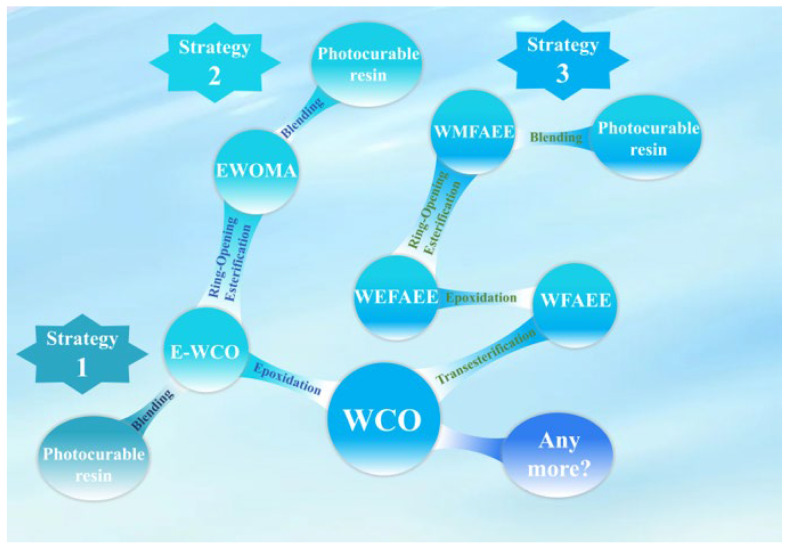
Synthesis strategies of WCO-derived photocurable resins.

**Figure 2 molecules-30-04000-f002:**
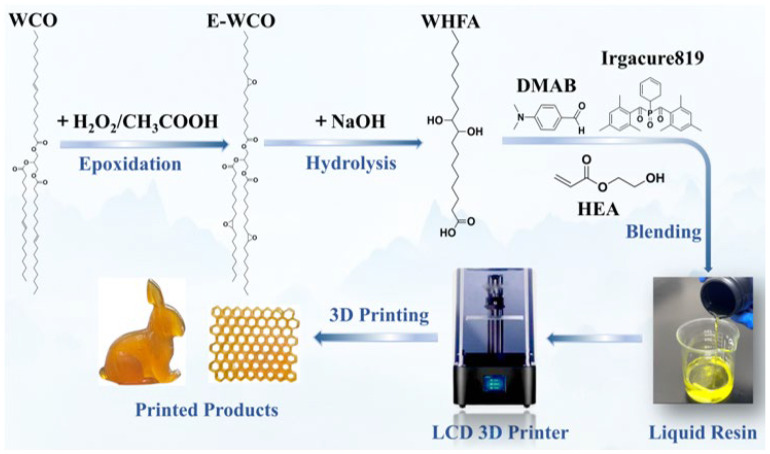
Schematic illustration of the preparation and 3D printing process of the WHFA/HEA photocurable elastomer.

**Figure 3 molecules-30-04000-f003:**
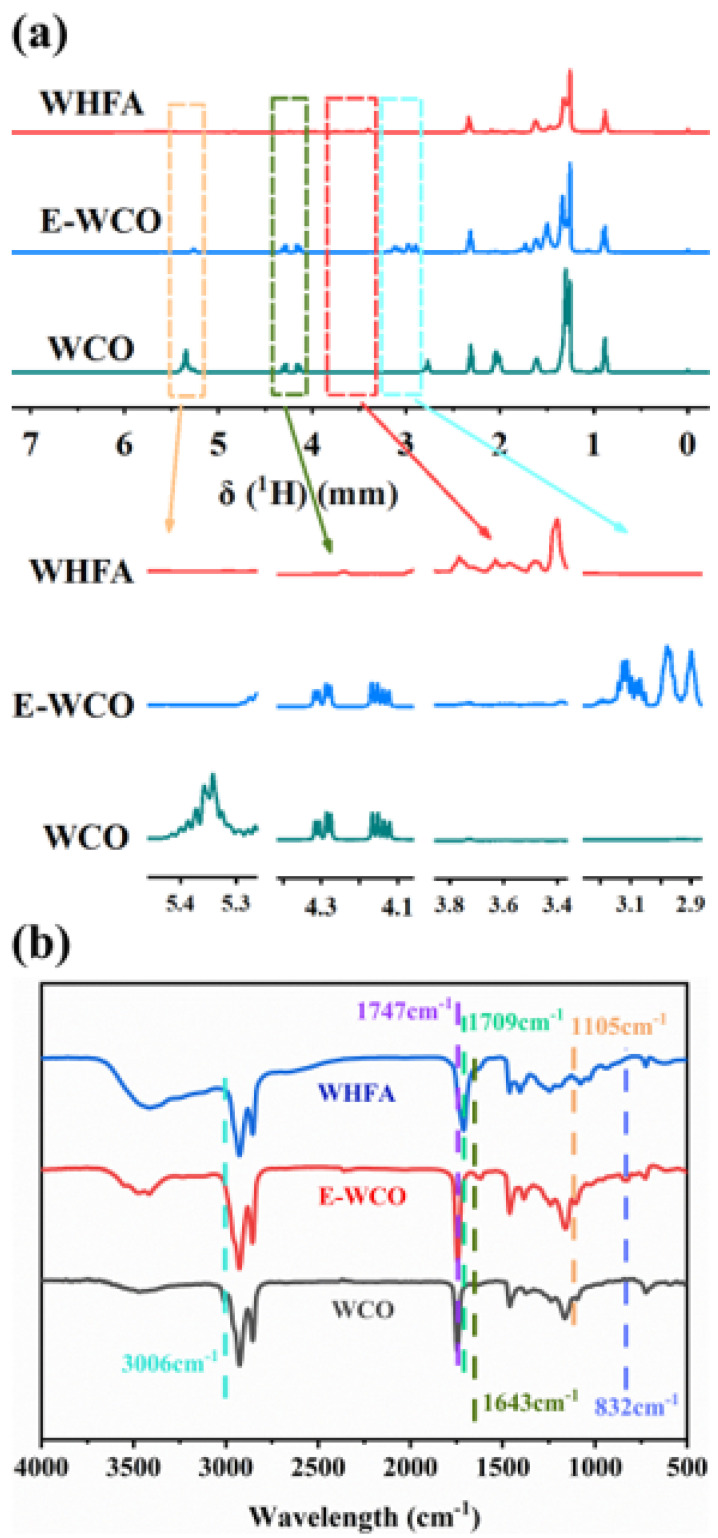
^1^H NMR (**a**) and IR spectra (**b**) of WCO, E-WCO and WHFA.

**Figure 4 molecules-30-04000-f004:**
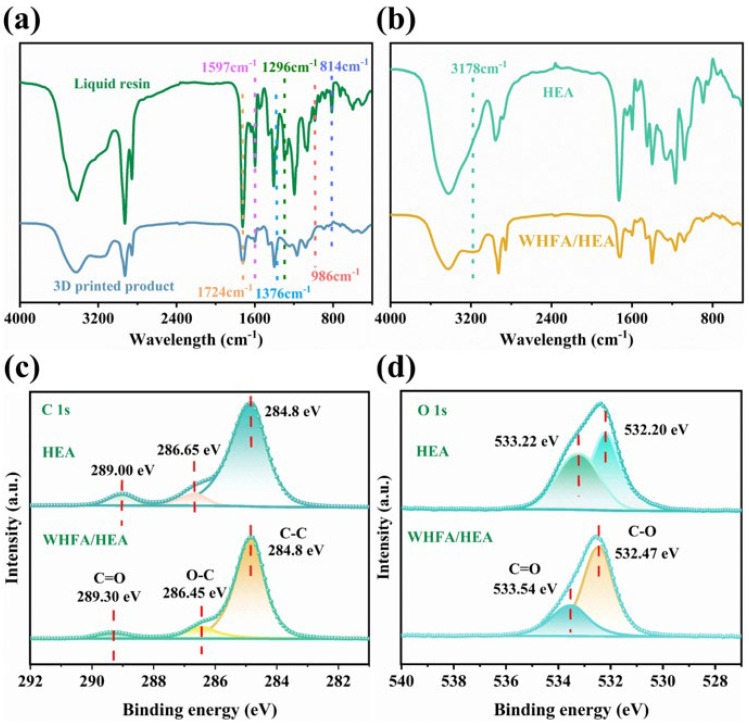
Spectroscopic analysis: (**a**) IR spectra of 3D printed product and liquid resin of WHFA/HEA (A2 sample); (**b**) IR spectra of 3D printed product of WHFA/HEA and pure HEA; (**c**–**d**) The C1s (**c**), and O1s (**d**) high-resolution XPS spectra of the WHFA/HEA resin.

**Figure 5 molecules-30-04000-f005:**
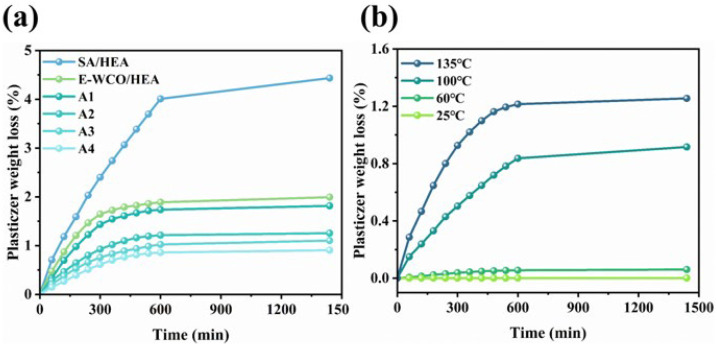
Migration behaviors: (**a**) Migration rates of WHFA in different WHFA/HEA resin systems, using E-WCO/HEA and SA/HEA resins as controls; (**b**) Migration rates of WHFA in the A2 resin system under varying temperature conditions.

**Figure 6 molecules-30-04000-f006:**
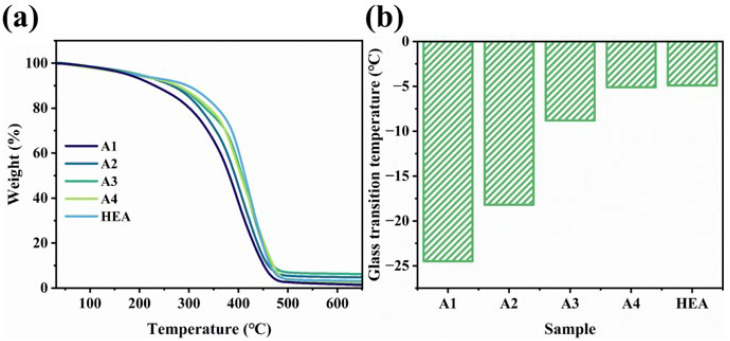
Thermal analysis: The TGA curves (**a**) and T_g_ values (**b**) of WHFA/HEA resins and pristine HEA.

**Figure 7 molecules-30-04000-f007:**
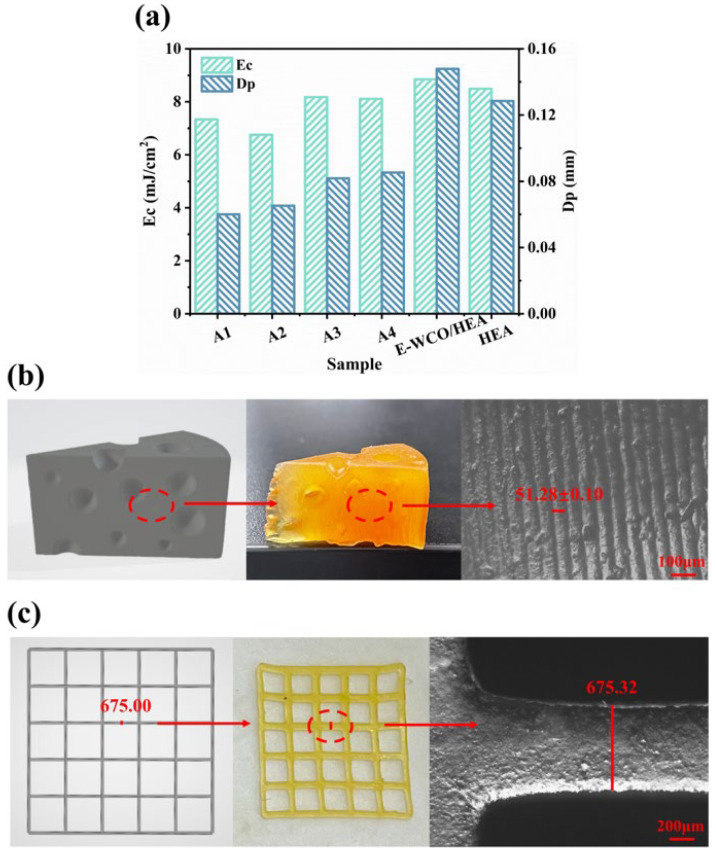
(**a**) Comparison of D_p_ and E_c_ values for WHFA/HEA resins with varying WHFA content, using E-WCO/HEA and pure HEA resins as controls; (**b**) CAD model of a cake structure (left), corresponding physical 3D-printed specimen (middle), and 100× magnified lateral-view metallographic micrograph (right); (**c**) CAD model of a lattice structure (left), physical 3D-printed lattice specimen (middle), and 50× magnified top-view metallographic micrograph (right).

**Figure 8 molecules-30-04000-f008:**
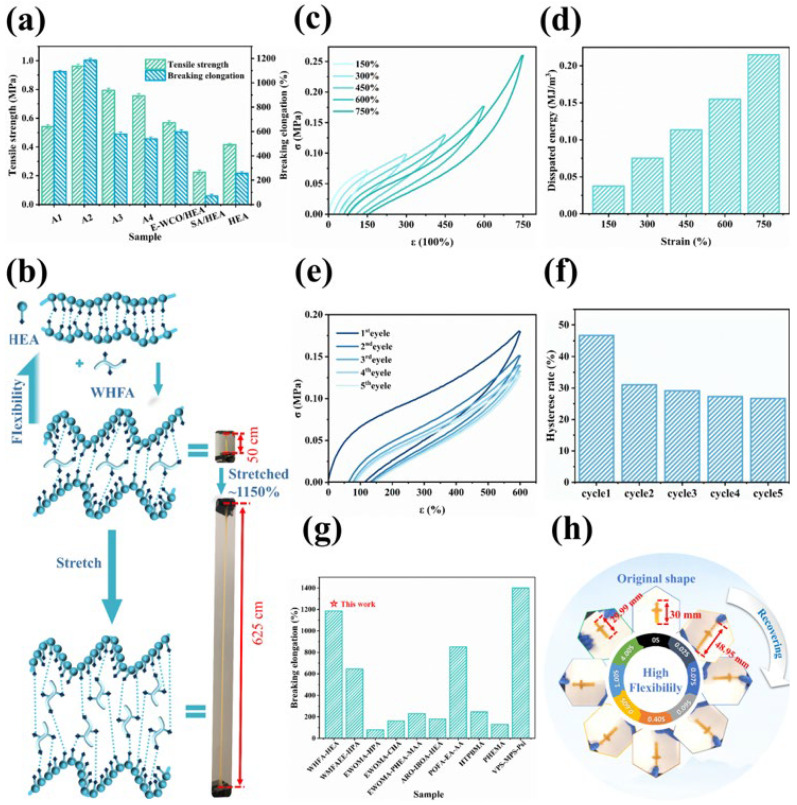
Mechanical properties of WHFA/HEA elastomer: (**a**) Tensile properties of WHFA/HEA resins with varying WHFA content; (**b**) A schematic illustration which depicted the structural evolution of WHFA/HEA elastomer during the tensile deformation, with cubic and pentagonal icons symbolizing hydroxyl (−OH) and carboxyl (−COOH) functional groups, respectively; (**c**) Stress–strain curves of A2 elastomer were plotted under varying strain levels; (**d**) Dissipated energy values of A2 elastomer were quantified across different strain magnitudes; (**e**) Cyclic stress–strain curves of A2 elastomer at 600% strain over five consecutive loading cycles; (**f**) HLR values of A2 elastomer at 600% strain over five consecutive loading cycles; (**g**) Comparison of the elongation at break of the WHFA/HEA elastomer with previously reported photocurable flexible resins; (**h**) Demonstration of elastic tensile deformation and cyclic recovery behavior in a 3D printed sword product fabricated from A2 elastomer.

**Figure 9 molecules-30-04000-f009:**
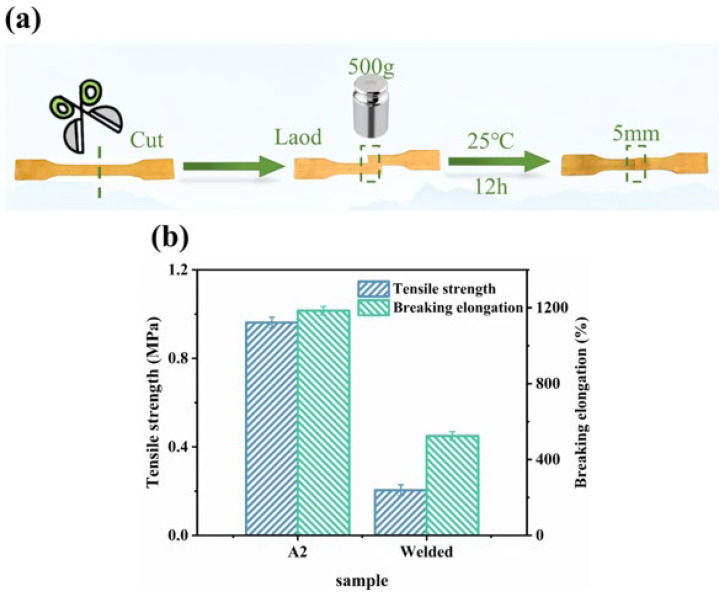
Welding performance of WHFA/HEA elastomer (A2 sample): (**a**) Photographs of tensile specimens captured prior to and following welding; (**b**) Mechanical property evaluation of A2 elastomer in both pre-welding and post-welding states.

**Figure 10 molecules-30-04000-f010:**
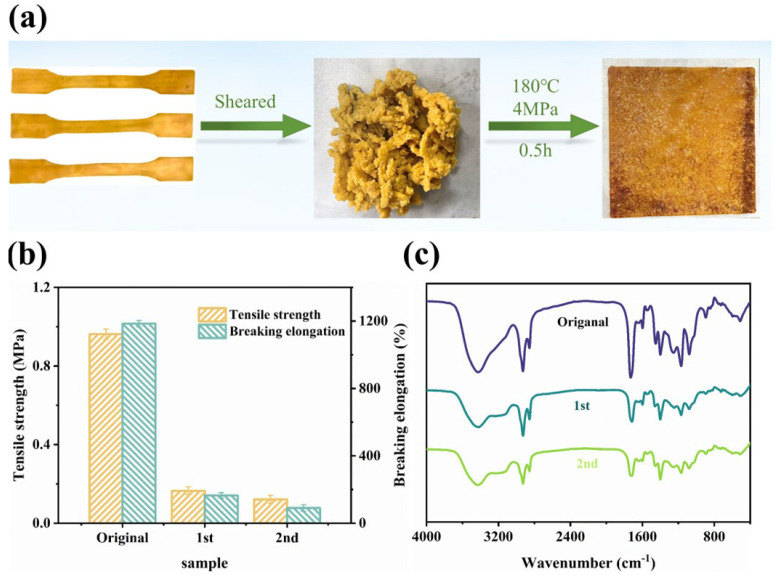
Physical reprocessability of WHFA/HEA elastomer (A2 sample): (**a**) Visual documentation of material fragments and thermally compressed sheets following reprocessing protocols; (**b**) Comparative analysis of mechanical performance metrics for A2 elastomer across two reprocessing cycles; (**c**) IR spectroscopic comparison between virgin and reprocessed elastomer samples.

**Figure 11 molecules-30-04000-f011:**
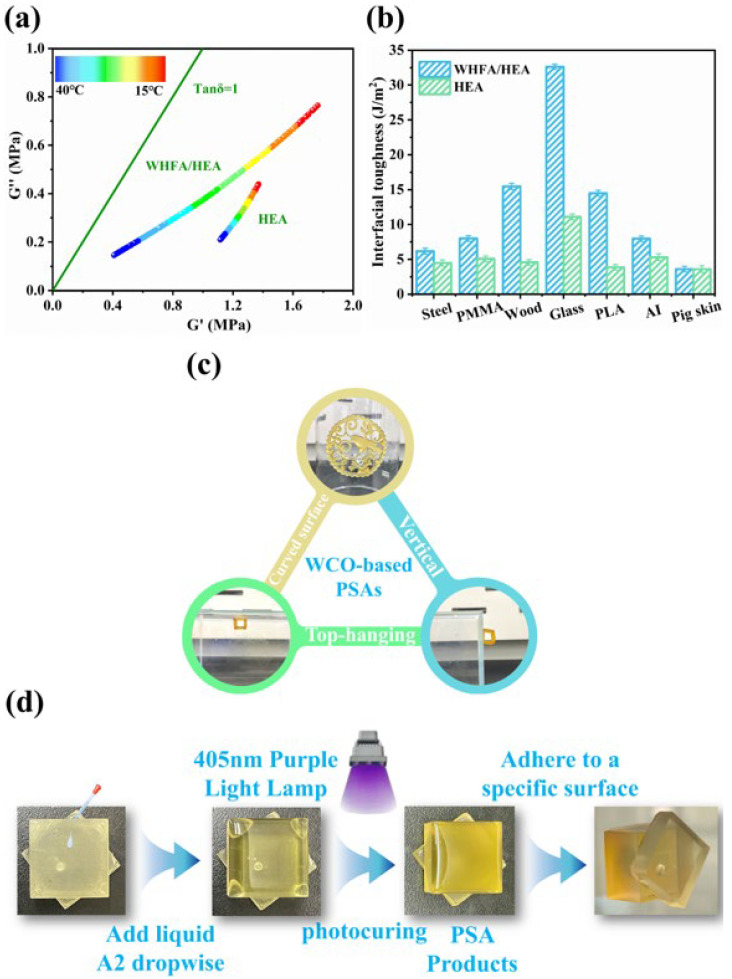
PSA properties of WHFA/HEA elastomer: (**a**) Viscoelastic response curves of pristine HEA and A2 elastomer samples measured across a temperature range of 15 °C to 40 °C; (**b**) Ambient-temperature interfacial adhesion toughness evaluation of pristine HEA and A2 elastomer on diverse substrate materials; (**c**) PSA performance demonstration using 3D printed products of A2 elastomer: (top) Hollowed-out circular pattern adhered to curved glass substrate; (left) Hollowed-out cubic structure suspended from overhead substrate; (right) Hollowed-out cubic structure vertically adhered to flat glass substrate; (**d**) Process illustration for manual fabrication of PSA products using WHFA/HEA elastomer. The cubic framework was 3D-printed with commercial resin, while the adhesive interface employed a coated, cured WHFA/HEA elastomer (A2 formulation) as the pressure-sensitive adhesive layer.

**Figure 12 molecules-30-04000-f012:**
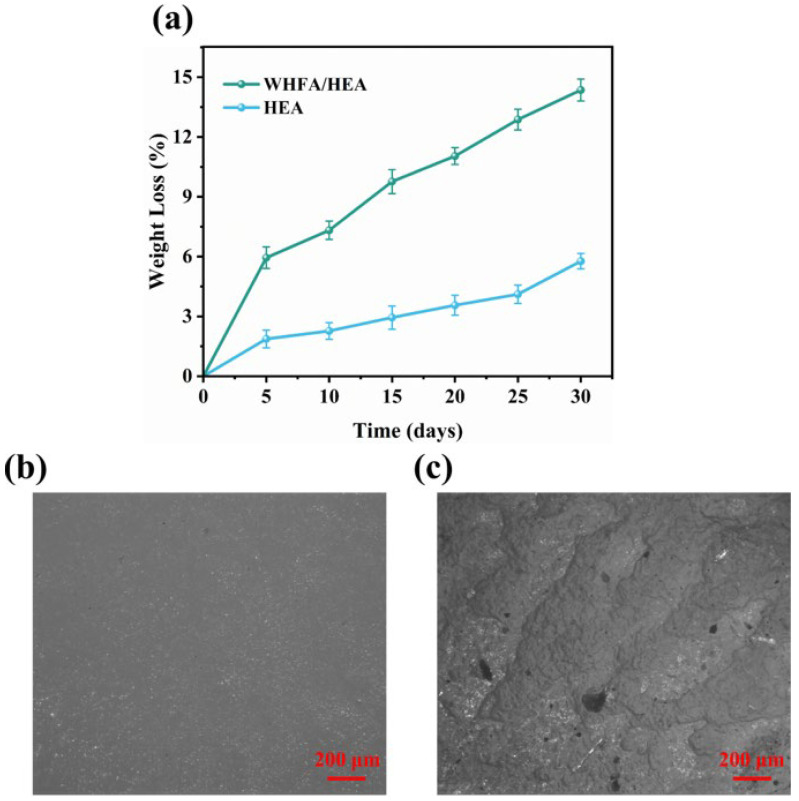
(**a**) Comparative biodegradation rates of the WHFA/HEA elastomer (A2 sample) and pure HEA during soil burial; (**b**,**c**) Surface microstructures of A2 sample before (**b**) and after (**c**) 45-day soil burial experiment.

**Table 1 molecules-30-04000-t001:** Comparison of D_p_ and E_c_ values for some reported 3D printing materials.

Major Compositions	D_p_ (mm)	E_c_ (mJ/cm^2^)	Reference
WHFA/HEA	0.065	6.75	This work
EWOMA -HPA	0.296	41.65	[48]
EWOMA -CHA	0.195	38.03	[48]
WMFAEE-HPA	0.224	55.44	[52]
Commercial resin	0.314	16.32	[69]
IPESO-ETPTA40	0.213	8.814	[8]
AESO	1.318	77.00	[70]
RSO-PUA/HEA40 %	0.327	15.20	[71]
CROSS−10	0.404	17.15	[72]
ChCl/HEMA/TA20	0.192	8.700	[73]

**Table 2 molecules-30-04000-t002:** Comparison of the mechanical performance of the WHFA/HEA elastomer with previously reported 3D printable photocurable flexible resins.

Major Compositions	Elongation at Break (%)	Tensile Strength (MPa)	Reference
Waste cooking oil-derived hydroxy fatty acid (WHFA) Hydroxyethyl acrylate (HEA)	1184.66	0.962	This work
WCO-based methacrylate fatty acid ethyl ester (WMFAEE) Hydroxypropyl acrylate (HPA)	645.09	0.967	[52]
Epoxy waste oil methacrylate (EWOMA) Hydroxypropyl acrylate (HPA)	78.34	0.368	[48]
Epoxy waste oil methacrylate (EWOMA) Cyclohexyl acrylate (CHA)	162.0	0.335	[48]
Epoxy waste oil methacrylate (EWOMA) 2-phenoxyethyl acrylate (PHEA) Methacrylic acid (MAA)	230.1	0.480	[50]
Acrylated rapeseed oil (ARO) Isobornyl acrylate (IBOA) Hydroxyethyl acrylate (HEA)	180	-	[15]
PO fatty acid-ethyl acrylamide (POFA-EA) Acrylic acid (AA)	851	4.2	[16]
Hydroxyl-terminated polybutadiene (HTPB) Hydroxypropyl methacrylate (HPMA)	246.1	32.9	[74]
Poly(2-hydroxyethyl methacrylate) (PHEMA)	130	25.4	[75]
Vinyl-terminated polydimethylsiloxane (VPS) Branched mercapto-functionalized polysiloxane (MPS) Precipitated silica (PSi)	1400	-	[76]

**Table 3 molecules-30-04000-t003:** Synthesis formulations of WHFA/HEA elastomers and their control samples.

Sample	WHFA (g)	HEA (g)	SA (g)	E-WCO (g)	Mass Ratio of HEA to WHFA (or SA, E-WCO)	Irgacure 819 (g)	DMAB (g)
A1	20	40	-	-	1:2	1.8	1.8
A2	15	45	-	-	1:3	1.8	1.8
A3	10	50	-	-	1:5	1.8	1.8
A4	10	100	-	-	1:10	3.3	3.3
SA/HEA (control sample)	-	45	15		1:3	1.8	1.8
E-WCO/HEA (control sample)	-	45	-	15	1:3	1.8	1.8
HEA (control sample)	-	100	-	-	-	3	3

## Data Availability

Data is contained within the article or Appendix A.

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
