# Peer review of "Three-Dimensional Printable Photocurable Elastomer Composed of Hydroxyethyl Acrylate and Hydroxy Fatty Acid Derived from Waste Cooking Oil: An Innovative Strategy for Sustainable, Highly Flexible Resin Development"

_molecules, 2025, doi:10.3390/molecules30194000_

Round 1

Reviewer 1 Report

Comments and Suggestions for Authors

In the present paper, Shen et al. described in a precise and deep manner their work on the development of flexible resins for VAT photopolymerization. Their approach is based on the valorization of waste cooking oil, representing a valuable strategy to improve circularity and to push waste-derived material applications to higher added value products. The state-of-the-art and the main issues of existing WCO valorisation methods are well presented in the introduction, but I recommend shortening the introduction since it could be a little bit dispersive for readers. The experimental work and results are clearly explained, and the figures are well organised.

In light if this, I strongly recommend the paper for publication in Molecules.

Reviewer 2 Report

Comments and Suggestions for Authors

Dear authors,

Your manuscript has great potential. However, it needs a major review to be suitable for publication. You need to add references in the results/discussion section in order to improve the scientific soudness of the manuscript. For instance, although it may be common regions/absorption bands (NMR/IR) for the authors, references are always welcome to compare and constrast with other results in the literature.

Also, I could not find Figure S8 in the supplementary materials. I suggest a thorough review to serach for small problems like this. 

Reviewer 3 Report

Comments and Suggestions for Authors

This article presents a sustainable and innovative method to synthesize a highly flexible photocurable elastomer by transforming waste cooking oil into hydroxy fatty acids, which are then blended with hydroxyethyl acrylate for 3D printing applications. The authors also demonstrate weldability, reprocessability, pressure-sensitive adhesion, biodegradability, and excellent print resolution. This is a strong, well-conducted study that makes a valuable contribution to sustainable polymer and 3D printing materials research. I recommend minor revisions to address clarity and consistency. The most critical concerns are outlined below:

  1. Explicitly discuss tensile strength vs flexibility trade-off.
  2. There is an inconsistency in the reported elongation recovery after welding. In the abstract, the recovery is stated as “19.97% elongation retention after 12 h at 25 °C”, while in the main text (Section 2.7), the reported value is approximately 44.23%. Please clarify which value is correct, ensure consistency between the abstract and results, and explain any differences if they refer to distinct experimental conditions or formulations.
  3. Provide info on reproducibility/error bars.
  4. Some long sentences could be split for easier reading.
  5. “Result and discussion” should be plural.
  6. Expand briefly on potential applications and cost benefit.

Reviewer 4 Report

Comments and Suggestions for Authors

In scope of statements within 2.4, it is strongly advised to explain why the Tg modified, and in particular explain modified polymer chain structure of both neat and recycled materials. This will strengthen the manuscript and clarify why WHFA can act as e.g. plasticizer.

In scope of 2.7, the welding methodology is not well explained. This must be added, and the mechanism also to be clarified. Identically, the reprocessing & compression must be explained in detail.

Also identify if wavelength as for photocuring process is identical for all type of samples, because this will affect the processing method, Cd, etc.

In scope of 2.9, why is this integrated in the manuscript? If this is needed, an in depth understanding of the mechanism must be integrated, which is now missing.

Comments on the Quality of English Language

The manuscript will need a detailed optimization of English language to improve reading quality.

Round 2

Reviewer 2 Report

Comments and Suggestions for Authors

Dear authors, congratulations on the great work. The revised manuscript improved its quality significantly.

Reviewer 4 Report

Comments and Suggestions for Authors

The reviewer thanks the authors for improved version.

Comments on the Quality of English Language

It is advised a detailed English review of final version.